# Anemia, micronutrient deficiencies, malaria, hemoglobinopathies and malnutrition in young children and non-pregnant women in Ghana: Findings from a national survey

Rita Wegmüller[1], Helena Bentil[2], James P. Wirth[1], Nicolai Petry[1], Sherry A. Tanumihardjo[3], Lindsay Allen[4], Thomas N. Williams[5], Lilian Selenje[6], Abraham Mahama[6], Esi Amoaful[7], Matilda Steiner-Asiedu[2], Seth Adu-Afarwuah[2], Fabian Rohner[1] *

1 GroundWork, Fläsch, Switzerland, 2 Department of Nutrition and Food Science, University of Ghana, Legon, Ghana, 3 Department of Nutritional Sciences, University of Wisconsin-Madison, Madison, Wisconsin, United States of America, 4 USDA/ARS Western Human Nutrition Research Center, Davis, California, United States of America, 5 KEMRI-Wellcome Trust Research Programme, Kilifi, Kenya, 6 UNICEF, Accra, Ghana, 7 Ghana Health Service, Accra, Ghana

* fabian@groundworkhealth.org

**Data Availability Statement:** The data underlying the results presented in the study are owned by

## Abstract

Nationally representative data on the micronutrient status of Ghanaian women and children are very scarce. We aimed to document the current national prevalence of micronutrient deficiencies, anemia, malaria, inflammation, α-thalassemia, sickle cell disease and trait, and under- and over-nutrition in Ghana. In 2017, a two-stage cross-sectional design was applied to enroll pre-school children (6–59 months) and non-pregnant women (15–49 years) from three strata in Ghana: Northern, Middle and Southern Belt. Household and individual questionnaire data were collected along with blood samples. In total, 2123 households completed the household interviews, 1165 children and 973 women provided blood samples. Nationally, 35.6% (95%CI: 31.7,39.6) of children had anemia, 21.5% (18.4,25.0) had iron deficiency, 12.2% (10.1,14.7) had iron deficiency anemia, and 20.8% (18.1,23.9) had vitamin A deficiency; 20.3%(15.2,26.6) tested positive for malaria, 13.9% (11.1,17.3) for sickle trait plus disease, and 30.7% (27.5,34.2) for α-thalassemia. Anemia and micronutrient deficiencies were more prevalent in rural areas, poor households and in the Northern Belt. Stunting and wasting affected 21.4% (18.0,25.2) and 7.0% (5.1,9.5) of children, respectively. Stunting was more common in rural areas and in poor households. Among non-pregnant women, 21.7% (18.7,25.1) were anemic, 13.7% (11.2,16.6) iron deficient, 8.9% (6.7,11.7) had iron deficiency anemia, and 1.5% (0.8,2.9) were vitamin A deficient, 53.8% (47.6,60.0) were folate deficient, and 6.9% (4.8,9.8) were vitamin B12 deficient. Malaria parasitemia in women [8.4% (5.7,12.2)] was lower than in children, but the prevalence of sickle cell disease or trait and α-thalassemia were similar. Overweight [24.7% (21.0,28.8)] and obesity [14.3% (11.5,17.7)] were more common in wealthier, older, and urban women. Our findings demonstrate that anemia and several micronutrient deficiencies are highly present in Ghana calling for the strengthening of Ghana's food fortification program while overweight and obesity in

UNICEF Ghana and the Ministry of Health Ghana and contain confidential, identifying information. Data are available from UNICEF Ghana (accra@unicef.orgs) for researchers who meet the criteria for access to confidential data. The authors had no special access to the data.

**Funding:** Funding for data collection, laboratory analysis, and reporting were provided by UNICEF and Canada's Ministry of Foreign Affairs, Trade and Development through a grant between UNICEF Ghana and the University of Ghana (#43210308). TNW was funded through a fellowship from the Wellcome Trust (#202800). Funding for the development of this manuscript was provided by GroundWork, and page charges were borne by UNICEF.

**Competing interests:** The authors have declared that no competing interests exist. The authors alone are responsible for the views expressed in this publication and they do not necessarily represent the decisions, policy or views of UNICEF. This does not alter our adherence to PLOS ONE policies on sharing data and materials.'

**Abbreviations:** AGP, α1-acid glycoprotein; CRP, C-reactive protein; DHS, Demographic and Health Survey; EA, enumeration area; ELISA, enzyme-linked immunosorbent assay; GHS, Ghana Health Services; GMS, Ghana Micronutrient Survey; GSS, Ghana Statistical Services; ID, iron deficiency; IDA, iron deficiency anemia; IYCF, infant and young child feeding; MUAC, mid-upper arm circumference; ODK, Open Data Kit; PSC, preschool-age children; PSU, primary sampling unit; RBP, retinol-binding protein; VAD, vitamin A deficiency; WHO, World Health Organization; WRA, (non-pregnant) women of reproductive age (15–49 years old).

women are constantly increasing and need to be addressed urgently through governmental policies and programs.

## Introduction

According to the Global Burden of Disease Study, 2.36 billion individuals were affected by anemia worldwide in 2015 [1]. According to Stevens *et al.*, the estimated anemia prevalence for young children, non-pregnant women and pregnant women (71%, 48% and 56%, respectively) is high in Central and West Africa [2]. Developing countries accounted for more than 89% of cases of anemia in 2013 with an estimated 11 million people affected in Ghana, which corresponded to over 40% of the population [1]. The causes of anemia are multifactorial, particularly in areas with a high burden of infectious diseases such as Ghana, and include iron deficiency, hemoglobinopathies, communicable diseases such as malaria and helminthiasis, deficiencies of other micronutrients, and inflammation [3,4]. However, only scarce and geographically limited data are available on micronutrient deficiencies in Ghana indicating a high prevalence of some micronutrient deficiencies, such as iron with a prevalence of about 45% and vitamin A with 26% in preschool children, while only 6% and 10% were affected by zinc and riboflavin deficiency, respectively [5,6]. No information for other micronutrients and inflammation could be identified. Further, no national assessment of the prevalence of the haemoglobin disorders sickle cell disease and trait or α-thalassemia has previously been conducted in Ghana.

To address micronutrient deficiencies, the government of Ghana has developed mandatory food fortification standards for iodine (50 ppm at retail level) in salt (in 1996), and iron (58.5 mg/kg), zinc (28.3 mg/kg), B-vitamins (folic acid (2.08 mg/kg), B12 (0.01 mg/kg), thiamine (8.4 mg/kg), riboflavin (4.5 mg/kg), niacin (59.0 mg/kg)) and vitamin A (2.0 mg/kg) in wheat flour (in 2010), and vitamin A in vegetable oil (in 2010) [7]. A nationally representative market survey in 2011 confirmed that 95% of vegetable oil was adequately fortified, while only 23% of wheat flour samples were adequately fortified [7]. A nationally representative household coverage survey found that about one third of salt samples were adequately iodized [8]. In addition to mandatory food fortification, the vitamin A supplementation program has been implemented over many years but its success has not been monitored regularly.

In many low- and middle-income countries obesity and diet-related chronic diseases commonly co-exist with undernutrition indicating the existence of the double burden of malnutrition [9]. In Ghana in 2014, 19% of children were stunted and 11% underweight while more than 40% of women were overweight or obese, indicating that the double burden of malnutrition might be common [9].

The lack of up-to-date nutrition-related data poses challenges for policy and program makers. The 2017 Ghana Micronutrient Survey (GMS), partly reported in this paper, was implemented to assess nutritional status measuring both micro- and macro-nutrient indicators in Ghana, as well as to gather information on other factors leading to malnutrition and anemia. In this paper, we present national and regional results for children and non-pregnant women. Although the prevalence of anemia and underweight in pregnant women was also measured in the GMS, these results are presented elsewhere [10].

## Methods

### Ethics and consent

Ethical approval for the survey protocol was obtained from the Ethics Review Committee of the Ghana Health Service (GHS), number GHS-ERC-15/01/2017. The survey protocol was

also registered with the Open Science Framework study registry (DOI: 10.17605/OSF.IO/J7BP9) [10].

Verbal informed consent was obtained for household interviews while written informed consent was sought from eligible non-pregnant women and from the caregivers of children eligible for individual interviews, anthropometry and blood sampling. For illiterate participants, the consenting procedure was done orally, followed by a fingerprint stamp and a witness signature as evidence of consent in lieu of a signature. Survey participants diagnosed with malaria, severe anemia, or severe acute malnutrition were referred to the nearest health facility for further testing and treatment.

## Study design and participants

An a *priori* sample size for the entire survey and each stratum was based on the estimated national prevalence, the desired precision around the resulting estimate of prevalence, and the expected design effect for priority indicators of nutritional status in children 6–59 months of age and non-pregnant women. Calculations assumed an expected household response rate of 92%, individual response rates for interview questions and anthropometric measurements of 95%, and response rates for venipuncture of 90% for women and 85% for children.

The survey used a two-stage cluster sampling with implicit stratification to select random samples of children aged 6–59 months and non-pregnant women of reproductive age (WRA). Sampling was done separately in three strata to represent areas with different agricultural and climatic conditions: 1) Southern Belt (Greater Accra, Central, Volta, Western regions); 2) Middle Belt (Brong-Ahafo, Ashanti, Eastern regions), and; 3) Northern Belt (Northern, Upper East, Upper West regions). Children and women met the inclusion criteria if they were in the appropriate age range, considered a member of the household, and written consent for participation was provided.

As a first stage, 30 census enumeration areas (EAs) within each of the three strata were randomly selected with probability proportional to population size. EAs were selected from a list of national EAs from Ghana's 2010 national census [11]. The field teams conducted a household listing exercise within selected EAs to generate updated household lists. For the second stage of sampling, a random selection of households in each EA was completed by using simple random sampling with equal probability of a specific number of households in each selected EA using a random number table. Because household size is quite different among regions, the number of households selected per EA depended on the region: 29 households per EA in the Southern and Middle strata, 20 households per EA in the Upper West and Upper East regions of the Northern stratum, and 15 households per EA in the Northern region of the Northern stratum. All children and pregnant women were recruited in selected households, while recruitment of non-pregnant women was done in every second selected household.

## Data collection procedures

Data collection for the GMS 2017 occurred between April and June 2017. Field data collection was preceded by a 10-day training and pre-testing on data collection procedures for supervisors, team leaders, interviewers, phlebotomists, anthropometrists, and laboratory technicians. Training included theory and role plays of different aspects of the survey, standardization exercises for anthropometrists for measuring height and weight, and for laboratory technicians for blood collection and processing, followed by a two-day pre-testing exercise of all survey procedures in two EAs (one rural and one urban) which were not part of the GMS 2017 sample. For this, all teams visited the same either rural or urban EA on the two subsequent days, and each

team was assigned 3–4 households to complete all steps. Field workers' performances were observed throughout and also assessed in a written test.

After having conducted the household listing exercise and random selection of households in each EA, the interviewers visited selected households. All questionnaires were written in English, but interviewers were assigned to teams according to language ability so that all interviews were conducted by a native speaker of the participant's language. Interviewers had 'cheat sheets' which contained translations of difficult or uncommon words and phrases in several languages, including Twi, Ga, Fante, Ewe, Dagaare, Waale, Frafra, and Dagbani.

Household interviews were conducted with the household head or another knowledgeable adult household member. Questions gathered data on household composition, household demographic characteristics, water and sanitation facilities, socio-economic status, and food purchasing patterns. Using the information on household composition, the tablet computers used to record interview data automatically identified which household members were eligible for recruitment (women 15–49 years of age in every second selected household and children 6–59 months of age in every selected household). After written consent was obtained, eligible women were interviewed about age, marital status, education, pregnancy and lactation status, antenatal care, dietary diversity, and consumption of vitamin and mineral supplements. The caregivers of eligible children were interviewed about child age, sex, recent morbidity, infant and young child feeding practices, consumption of micronutrient-rich foods or supplements, and vitamin A supplementation. The household, child, and woman questionnaire data were all collected electronically using tablet computers and the Open Data Kit (ODK) software.

Upon completion of individual questionnaires, the interviewers completed a paper form that invited the participant to go to a central place in the EA for anthropometry and blood sampling. Weight (using a Seca scale model 877, Hamburg, Germany) and height or length (using a height board from UNICEF item number S0114540, Copenhagen, Denmark) were measured according to standard procedures for both children and women [12]. Scales were quality controlled on a daily basis using calibration weights. For children who could not stand by themselves, the mother or caregiver was first weighed alone, then together with the child and the child weight indicated by using the scale's tare function. For children under two years, measurement of recumbent length was taken. The feet of children were examined for edema, and edema was only considered present if it was bilateral.

Subsequently, a 300 µL capillary blood sample was obtained from most children by finger prick into a silica-coated container (Sarstedt Microvette, Nümbrecht, Germany). From approximately 15% of all children, venous blood was collected to measure serum retinol concentration. From non-pregnant women, 6 mL of venous blood were collected by venipuncture into a "red top" silica-coated tube (Becton Dickinson Vacutainer, Franklin Lakes, NJ, USA).

Hemoglobin concentration was measured using a portable hemoglobinometer (Hb301, HemoCue AB, Ängelholm, Sweden) and malaria status (current or recent parasitemia) was checked using the SD BIOLINE Malaria Ag Pf/Pan rapid diagnostic test kit (Standard Diagnostics Inc, Gyeonggi-do, Republic of Korea), which distinguishes between infection with *P. falciparum* and infection with another *Plasmodium* species (*P. vivax*, *P. malariae or P. ovale*). Daily quality control of the HemoCue devices was conducted using liquid control blood (Eurotrol, Ede, the Netherlands). In children, the 3rd and 4th drops of blood were used to measure hemoglobin concentration and malaria status, respectively. In women and in the 15% children for whom venous blood was collected, two drops of venous blood for hemoglobin and malaria testing were extracted from the blood tube into weighing boats using a DIFF-Safe adaptor (Becton Dickinson, Franklin Lakes, NJ, USA).

The remaining blood from children and women was stored in a cold box and transported cold (between 1–8˚C) and protected from light to a laboratory place. Later on the day of

collection, samples were centrifuged and serum and pellet aliquots frozen at -20˚C. This temperature was maintained during domestic transport and sample sorting. Samples were packed on dry ice during shipment to international laboratories for further analysis.

Serum samples of all children and women were analyzed at the VitMin-Lab (Willstaett, Germany) for retinol-binding protein (RBP), ferritin, C-reactive protein (CRP) and α1-acid glycoprotein (AGP) using a sandwich ELISA [13]. Because RBP is not yet recommended by WHO as a standard test for vitamin A status, 300 samples were measured for serum retinol concentration using high-performance liquid chromatography at the University of Wisconsin. The Pearson correlation between these two test results is of 0.9039 and the linear regression equation is RBP = 1.1486 x retinol—0.016.

Additionally, serum aliquots for women were analyzed at the USDA/ARS Western Human Nutrition Research Center (Davis, USA) for folate and vitamin B12 concentrations using the Cobas e411 analyzer (Roche Diagnostics, Indianapolis, USA).

Pellets from all children and from a randomly selected sub-sample of approximately one-half of the women were tested for sickle cell disease and trait and α-thalassemia at the laboratories of the KEMRI-Wellcome Trust Research Programme, Kilifi, Kenya, by using polymerase chain reactions [14–16].

All the laboratories participate in external quality control schemes and have established internal quality control mechanisms. In addition, laboratories successfully participating in CDC's VITAL-EQA program demonstrate acceptable performance in measuring RBP, ferritin, CRP, AGP, retinol, folate and vitamin B12.

## Case definitions

Children with z-scores below -2.0 for weight-for-height and height-for-age were classified as wasted or stunted, respectively [17]. Moderate wasting and stunting were defined as a z-score less than -2.0 but greater than or equal to -3.0, and severe wasting and stunting were denoted by z-scores less than -3.0. Bilateral pitting edema in the feet and/or lower legs automatically qualified for severe wasting, regardless of their weight-for-height z-score. Overweight was defined as a weight-for-height z-score of greater than +2.0 but less than or equal to +3.0 and obesity as a weight-for-height z-score greater than +3.0. For women, the following cut-off points for body mass index (BMI) to define levels of malnutrition in non-pregnant women were used: <16.0 severe undernutrition, 16.0–16.9 moderate undernutrition, 17.0–18.4 at risk of undernutrition, 18.5–24.9 normal, 25.0–30.0 overweight and >30.0 obese [18].

Anemia was defined according to WHO recommendations and was adjusted for smoking, where applicable [19]. For preschool-age children (PSC), the following anemia cutoffs were used: <70 g/L severe, 70–99 g/L moderate, 100–109 g/L mild; For WRA, the thresholds are: <80 g/L for severe anemia, 80–109 g/L for moderate anemia, and 110–119 g/L for mild anemia. For both PSC and WRA, ferritin and RBP concentrations were adjusted for elevated AGP and CRP according to the procedure recommended by Thurnham [20]. In children, iron deficiency was defined as inflammation-adjusted serum ferritin <12 μg/L, while in women it was <15 μg/L [21]. Iron deficiency anemia was defined as anemia with concomitant iron deficiency; non-anemic children for whom iron status results were missing were coded as *not* having iron deficiency anemia (n = 7). RBP concentrations were adjusted for elevated AGP and CRP [22], and vitamin A deficiency in children and women was defined as RBP <0.7 μmol/L [23]. Folate deficiency was defined as plasma folate <10 nmol/L, and plasma vitamin B12 concentrations <148 pmol/L were classified as "deficient" and 148 pmol/L to 220 pmol/L as "marginal" [24]. Inflammation status was classified into four categories: no inflammation with normal CRP and AGP, incubation with elevated CRP (>5mg/L) alone, early convalescence

with elevated CRP and AGP (>1.0 g/L), and late convalescence with elevated AGP alone. In addition, elevated CRP and/or AGP were used to identify individuals as having *any* inflammation.

The proportion of women consuming a minimum acceptable diet was calculated according to the 'Food and Nutrition Technical Assistance' guidelines [25]. Infant and Young Child Feeding (IYCF) indicators were calculated according to WHO/UNICEF guidelines [26].

### Data management and statistical analysis

Direct electronic data entry was done by using ODK during household, child, and woman interviews. The results of anthropometry and blood testing which were initially recorded on paper forms were entered into ODK on the same or the following day. Team leaders regularly cross-checked entries prior to uploading them, and once uploaded, remote quality control was done by the research team. Upon completion of fieldwork, databases were merged, with laboratory data either been auto-generated or manually entered (Microsoft Excel, version 2013). Data analysis was done using Stata/IC version 14.2.

Analyses were weighted to account for the unequal probability of selection in the three strata and the inaccurate estimates of primary sampling unit (PSU) size used during stage one sampling. The actual probability of selection for each selected EA during stage one sampling was calculated using 2010 census data. A corrected probability of selection was calculated using the actual number of households listed in each selected EA during the GMS household listing and the 2017 estimated population of each stratum. These two probabilities were then used to calculate a standardized sampling weight to correct for inaccurate estimates of PSU size. The standardized stratum-specific and PSU-specific sampling weights were then multiplied to produce an overall standardized sampling weight for each EA. This combined weight was used for all analyses except those in which stratum was an independent variable; for these analyses, only the PSU-specific weight was used. The statistical precision of all prevalence and mean estimates were assessed using 95% confidence limits, which were calculated accounting for the complex sampling. For dichotomous or categorical data, the chi-square test was applied. To compare the difference between means the t-test and between medians the Mann-Whitney test were used.

Using data on each household's dwelling, water and sanitation conditions and facilities, and ownership of durable goods, a wealth index was calculated by using the method described by the World Bank [27]. Calculation of wealth index quintiles categorizes the continuous wealth index and permits the cross-tabulation and the subsequent presentation of key indicators by wealth quintile. WHO/UNICEF classifications were used to define household water sources as "safe" or "unsafe" and sanitation facilities as "improved" or "unimproved" [28].

Geographic analysis techniques were employed using the interpolation function of Quantum GIS 3.4 to present the geographic distribution of nutrition outcomes. Specific estimates for selected EAs of a given prevalence were linked to latitude and longitude coordinates, and inverse distance weighting was used to estimate these distributions [29]. EAs with fewer than 5 subjects for the specific indicator were set to missing prior to running the interpolation. For the weighting procedure, a distance coefficient P, which specifies the degree of influence as distance from the point increases, was set to 2 due to the relatively consistent and small distance between many of the EAs.

## Results

### Response rates and household demographics

Overall, 2123 (98.3%) of 2159 selected households consented to and participated in the household questionnaire data collection. Only 1.0% of 1064 eligible women in participating

households refused or were unavailable to participate in the interview, and 91.4% consented to blood collection. None of the caretakers of the 1232 eligible children refused interview, but 50 (4.1%) refused anthropometry and phlebotomy on their child. Blood was successfully collected from 94.4% of eligible child participants.

Of the households selected, 52% were located in urban areas, comparable to the 51% found in the 2010 census [11]; regional proportions were rather comparable to the 2010 census (Southern Belt 41.1 in 2017 GMS vs. 43.4% in 2010 census; Middle Belt 43.0 vs. 39.5%; Northern Belt 15.8 vs. 17.1%). Mean household size was 4.2 (95% CI: 4.0, 4.4) and 75% of household heads reported having attended school or pre-school. Natural cooking fuels such as wood and charcoal were used in 77% of the households. Inadequate sanitation was found in 87% of the households while 91% regularly drank "safe" water.

## Children

Child characteristics and information on recent or current illness and infection as well as of some IYCF indicators are shown in Table 1. About one-half of the children were male, and almost one-half lived in urban households. IYCF indicators for breastfeeding and introduction of complementary food were high, whereas only 14.3% of the children met a minimum acceptable diet the day before the interview.

Illnesses in the past two weeks prior to the survey were common with close to one-quarter of children reporting diarrhea and one-quarter with a cough. Almost one-third of children had a caregiver-reported fever. Almost one-half of the children had some inflammation.

About one-fifth of children had malaria parasitemia; 15.9% had *P. falciparum* only, 0.5% had *P. vivax* or *P. malariae* or *P. ovale* infections and 3.9% had infection with both *P. falciparum* and another *Plasmodium* species. Notably, malaria prevalence increased with age group from 9.7% in 6–11 month old children to 29.2% in 48–59 month old children (p<0.005). A higher proportion of male children (23.5% vs. 17.2% females, p<0.05), of children living in rural areas (29.3% vs. 8.0% in urban areas, p<0.001) and of children living in less wealthy households (over 20% in the three lowest vs. below 5% in the two highest wealth quintiles, p<0.001) were infected with malaria.

The prevalence of sickle cell disease (HbSS) in children was 1.3%, whereas the proportion of children with sickle cell trait (i.e. HbAS; carrying only one abnormal allele of the hemoglobin β gene) was 12.6%. Similarly, the prevalence of homozygous α-thalassemia was low (3.3%), but almost one third had heterozygous α-thalassemia.

Table 2 shows the results on nutritional and micronutrient status among children nationally, and by urban and rural stratification. Child stunting was common in Ghana with a total prevalence of 21.4% including 7.2% of severe stunting. The proportion of stunted children was lower in urban (13.5% vs. 27.1% in rural households, p<0.001) and in the wealthiest household quintile (7.6% vs. 18.3–25.3% in all other quintiles, p<0.005). Child wasting was relatively high (7.1%), but with only 1.9% of them suffering from severe acute malnutrition. The prevalence of wasting decreased with older age (12.6% in 6–11 to 5.1% in 48–59 month old children, p<0.01). Underweight in Ghanaian PSC was common with a prevalence over 15%. Wealthier children (6.1% in highest to 18.6% in lowest wealth quintile, P<0.05) and female children (12.7% vs. 17.6% in males, p<0.05) were less likely to be underweight.

Mean hemoglobin concentration among Ghanaian PSC was 112.4 g/L and over one third of children were anemic, but only 0.7% had severe anemia. Anemia prevalence was higher in rural areas (42.1% vs. 26.8% in urban areas, p<0.001) and in the Northern Belt (53.2% vs. 32.3% and 28.2% in the Southern and Middle Belt, respectively, p<0.001), and decreased with

**Table 1. Demographics, IYCF indicators, recent or current illness, infection and hemoglobinopathies among pre-school-age children, national, Ghana 2017.**

| Characteristic | n [a] | % [b] | (95% CI) [c] |
|---|---|---|---|
| Age Group (in months) | | | |
| 6–11 | 125 | 10.0 | (8.2; 12.1) |
| 12–23 | 292 | 24.1 | (21.1; 27.4) |
| 24–35 | 279 | 23.1 | (20.8; 25.6) |
| 36–47 | 270 | 21.8 | (19.4; 24.5) |
| 48–59 | 266 | 20.8 | (18.8; 22.9) |
| Male sex | 615 | 50.3 | (47.1; 53.6) |
| Residing in urban household | 465 | 43.7 | (31.9; 56.3) |
| Early initiation of breastfeeding [d] | 275 | 78.6 | (72.6; 83.5 |
| Continued breastfeeding at 1 year [e] | 93 | 93.1 | (81.9; 97.6) |
| Introduction of solid, semi-solid or soft foods [f] | 53 | 94.7 | (83.8; 98.4) |
| Minimum acceptable diet [g] | 54 | 14.3 | (10.8; 18.6) |
| Diarrhea in the past 2 weeks | 294 | 22.9 | (20.2; 25.8) |
| Fever in the past 2 weeks | 409 | 32.1 | (27.9; 36.6) |
| Illness with a cough in the past 2 weeks | 319 | 25.8 | (21.9; 30.2) |
| Any inflammation [h] | 521 | 46.0 | (40.0; 52.2) |
| None | 644 | 54.0 | (47.8; 60.0) |
| Incubation (elevated CRP only) | 39 | 2.8 | (1.9; 4.0) |
| Early convalescence (elevated CRP and AGP) | 199 | 17.7 | (14.1; 22.0) |
| Late convalescence (elevated AGP only) | 283 | 25.5 | (22.1; 29.3) |
| Malaria parasitemia | 233 | 20.3 | (15.2; 26.6) |
| Sickle cell disease or trait | 143 | 13.9 | (11.1; 17.3) |
| HbSS (disease) | 10 | 1.3 | (0.7; 2.3) |
| HbAS (trait) | 133 | 12.6 | (10.0; 15.8) |
| α-thalassemia (combined) | 350 | 30.7 | (27.5; 34.2) |
| Homozygous | 35 | 3.3 | (2.3; 4.7) |
| Heterozygous | 315 | 27.4 | (24.4; 30.7) |

[a] Total n was 1232 for questionnaire-related variables, 1184 for anthropometric measures, and between 1113 to 1165 for blood biomarkers. The n's are un-weighted nominators in each subgroup; the sum of subgroups may not equal the total because of missing data.

[b] Percentages weighted for unequal probability of selection.

[c] CI = confidence interval, calculated taking into account the complex sampling design.

[d] Results presented for children 6–24 months of age; assessed as 'put to breast within 1 h of birth'.

[e] Results presented for children 12–15 months of age; assessed as 'breastfed the day before interview'.

[f] Results presented for children 6–8 months of age; assessed as 'received solid, semi-solid or soft foods the day before the interview'.

[g] Results presented for children 6–23 months of age; assessed the day before interview using a combination of minimum dietary diversity and minimum meal frequency.

[h] CRP = C-reactive protein, AGP = α1-acid-glycoprotein.

older age (46.1% in 12–23 to 23.4% in 48–59 months old children, p<0.001) and with higher wealth quintile (47.0% in lowest to 13.8% in highest, p<0.001).

Median ferritin concentration was 25.3 µg/L (IQR: 12.4, 44.3), resulting in 21.5% of PSC with iron deficiency (ID) and 12.2% with iron deficiency anemia (IDA). Similar to anemia, we found the highest prevalence of ID (34.4%) and IDA (22.1%) in 12–23 months old children with a progressive decreased with increasing age (10.2% for ID and 5.0% for IDA at 48–59

**Table 2. Nutritional and micronutrient status among preschool-age children, Ghana 2017.**

| Characteristic | National | | | Urban | | | Rural | | | p-value [c] |
|---|---|---|---|---|---|---|---|---|---|---|
| | N | % / Mean [a] | 95%CI [b] | N | %/ Mean | 95%CI | N | %/ Mean | 95%CI | |
| **Nutritional status** | | | | | | | | | | |
| Stunting | | | | | | | | | | |
| HAZ, mean | 1163 | -0.96 | (-1.09; -0.82) | 426 | -0.62 | (-0.82; -0.42) | 737 | -1.21 | (-1.34; -1.08) | <0.001 |
| Any stunting, % | 1163 | 21.4 | (18.0; 25.2) | 426 | 13.5 | (9.4; 19.0) | 737 | 27.1 | (23.2; 31.4) | <0.001 |
| Severe stunting, % | 1163 | 7.2 | (5.4; 9.7) | 426 | 3.8 | (2.0; 7.3) | 737 | 9.7 | (7.3; 12.9) | <0.01 |
| Wasting, overweight and obesity | | | | | | | | | | |
| WHZ, mean | 1156 | -0.52 | (-0.60; -0.45) | 423 | -0.51 | (-0.63; -0.39) | 733 | -0.53 | (-0.63; -0.43) | 0.82 |
| Any wasting, % | 1156 | 7.0 | (5.1; 9.5) | 423 | 7.6 | (4.9; 11.7) | 733 | 6.5 | (4.2; 10.0) | 0.62 |
| Severe wasting, % | 1156 | 1.9 | (1.3; 2.9) | 423 | 2.4 | (1.4; 4.1) | 733 | 1.5 | (0.8; 2.8) | 0.28 |
| Overweight/Obesity, % | 1156 | 0.7 | (0.3; 1.5) | 423 | 0.8 | (0.3; 2.5) | 733 | 0.6 | (0.2; 1.8) | 0.68 |
| Underweight | | | | | | | | | | |
| WAZ, mean | 1165 | -0.91 | (-1.01; -0.81) | 428 | -0.71 | (-0.88; -0.55) | 737 | -1.05 | (-1.15; -0.94) | <0.01 |
| Any underweight, % | 1165 | 15.8 | (13.2; 18.8) | 428 | 13.2 | (9.4; 18.3) | 737 | 17.7 | (14.5; 21.5) | 0.14 |
| Severe underweight, % | 1165 | 4.3 | (3.2; 5.9) | 428 | 3.9 | (2.1; 7.1) | 737 | 4.7 | (3.4; 6.4) | 0.59 |
| **Micronutrient status** | | | | | | | | | | |
| Hemoglobin concentration | | | | | | | | | | |
| Hemoglobin (g/L), mean | 1172 | 112.4 | (110.9; 113. 9) | 432 | 115.4 | (112.8; 117.9) | 740 | 110.2 | (108.6; 111.8) | <0.01 |
| Any anemia, % | 1172 | 35.6 | (31.7; 39.6) | 432 | 26.8 | (21.1; 33.4) | 740 | 42.1 | (37.3; 47.0) | <0.001 |
| Moderate anemia, % | 1172 | 17.0 | (14.0; 20.6) | 432 | 12.5 | (8.5; 17.9) | 740 | 20.4 | (16.3; 25.2) | <0.05 |
| Severe anemia, % | 1172 | 0.7 | (0.3; 1.5) | 432 | 0.1 | (0.0; 0.8) | 740 | 1.2 | (0.5; 2.5) | <0.01 |
| Iron status | | | | | | | | | | |
| Ferritin (μg/L), median (IQR) [d] | 1165 | 25.3 | (12.4; 44.3) | 426 | 22.5 | (12.4; 38.6) | 739 | 27.6 | (12.4; 28.6) | <0.01 |
| Iron deficiency, % [e] | 1165 | 21.5 | (18.4; 25.0) | 426 | 21.2 | (16.8; 26.5) | 739 | 21.8 | (17.1; 27.3) | 0.89 |
| Iron deficiency anemia, % | 1165 | 12.2 | (10.1; 14.7) | 429 | 10.0 | (7.0; 14.0) | 737 | 13.9 | (10.6; 18.0) | 0.1684 |
| Vitamin A status | | | | | | | | | | |
| RBP (μmol/L), mean [f] | 1165 | 0.93 | (0.91; 0.96) | 426 | 0.95 | (0.91; 0.99) | 739 | 0.92 | (0.88; 0.95) | 0.22 |
| Vitamin A deficiency, % [g] | 1165 | 20.8 | (18.1; 23.9) | 426 | 18.0 | (13.4; 23.9) | 739 | 22.9 | (19.7; 26.4) | 0.149 |

[a] Percentages/means weighted for unequal probability of selection.

[b] CI = confidence interval, calculated taking into account the complex sampling design.

[c] P-values measuring the differences in mean and prevalence between urban and rural areas was calculated using t-test and chi-square test; the Mann-Whitney test was used to calculate p-values between medians.

[d] Adjusted for inflammation [20]; corresponding unadjusted median ferritin concentration 31.4 μg/L (IQR: 15.4; 60.0).

[e] Based on inflammation-adjusted ferritin concentration [20].

[f] Retinol-binding protein, adjusted for inflammation [22]; corresponding unadjusted mean RBP concentration 0.87 μmol/L (95%CI: 0.84; 0.89).

[g] Based on the retinol-binding protein, adjusted for inflammation [22].

months of age). The prevalence of ID and IDA was markedly higher among children residing in the Northern Belt (39.6 and 29.0%) compared to the Middle (17.8 and 7.9%) and Southern Belts (12.6 and 5.2%) (p<0.0001). IDA prevalence decreased with increasing wealth quintile (from 20.0 to 5.2, p<0.001).

Nationally, about one-fifth of children had vitamin A deficiency (VAD) using inflammation-adjusted RBP as the indicator. Children living in the Northern Belt had a higher VAD prevalence (30.6% vs. 17.0% in the Southern and 18.1% in the Middle Belt, p<0.001) and prevalence decreased with increasing wealth quintile from 25.1% in the lowest to 9.1% in the highest quintile (p = 0.014).

As shown in the geospatial maps (Fig 1), the prevalence of anemia (Fig 1B), ID (Fig 1C), and VAD (Fig 1D) were markedly higher in the north of the country (i.e. Upper West, Upper East, Northern). Iron deficiency was very rare in the southern regions while anemia was common. In addition to these general geographic trends, there were "pockets" within the country where prevalence was higher than found in other areas of the region, which is clearly shown for stunting (Fig 1A) with an overall low prevalence but a few pockets where >40% of children were stunted.

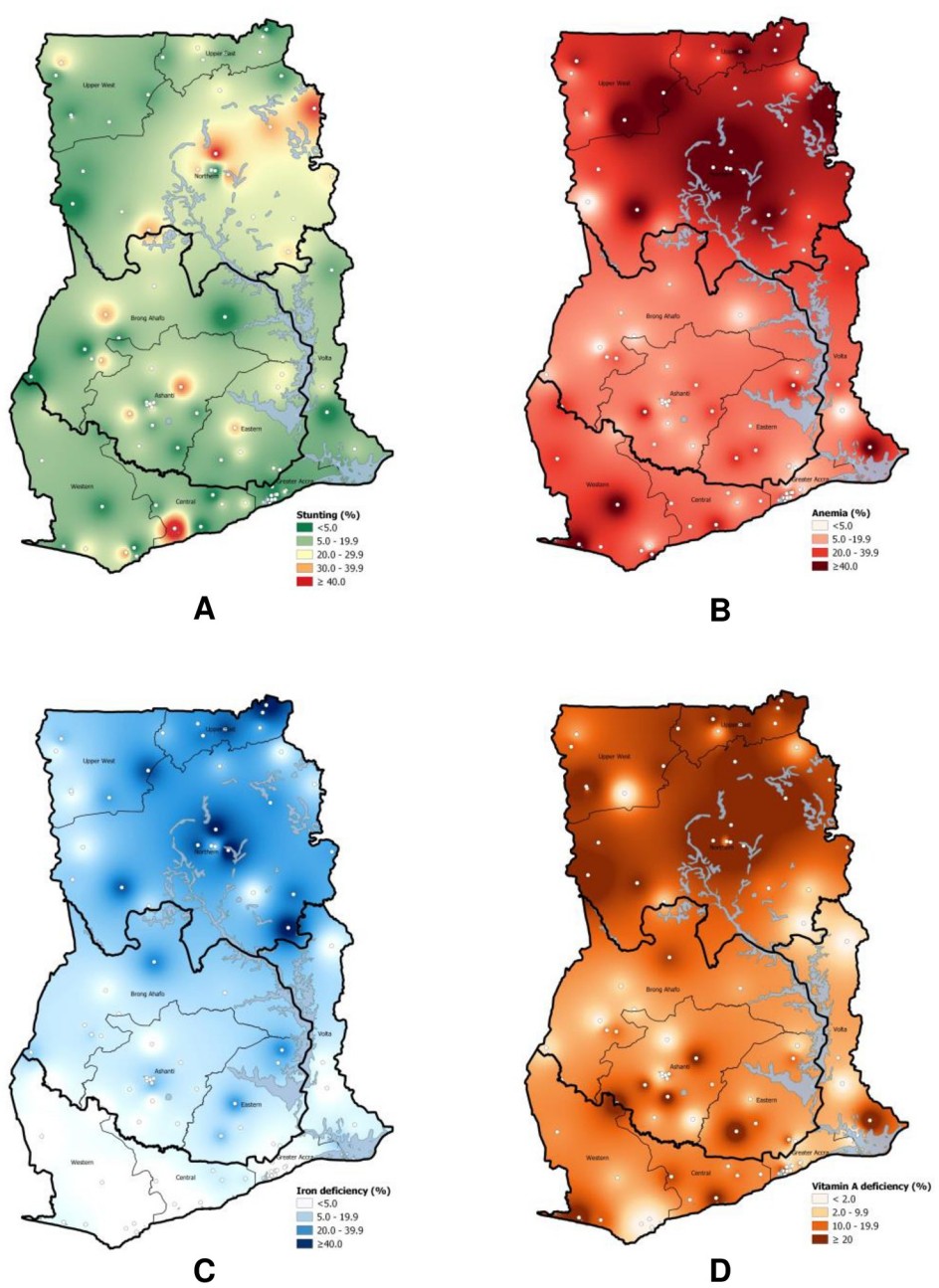

**Fig 1.** Geospatial distribution of prevalence of stunting (A), anemia (B), iron deficiency (C) and vitamin A deficiency (D) among preschool-age children, Ghana. White dots represent the EAs.

## Non-pregnant women of reproductive age

Mean age (SD) of WRA enrolled into the survey was 29.1 (±9.2) years, and 18.7% (95% CI: 15.8, 21.9) of all non-pregnant women were lactating at the time of the survey. The majority of the women were either married (59.9%) or had never married/lived with a man (30.7%). While more than half of the women had no job outside the home, almost one-third had a skilled or professional job.

Table 3 summarizes additional data on women demographics, diet, infection and hemoglobinopathies. Participating women were equally distributed between rural and urban areas and almost half reached a minimum dietary diversity score. Almost one-fifth of the women never attended school, and more than one-quarter were illiterate. Nearly 80% of women did not have inflammation at the time of the survey. Those with inflammation were equally distributed among the incubation, early convalescence, and late convalescence categories. Inflammation prevalence was significantly (p<0.05) higher in malaria positive women (39.3%) than in malaria negative women (19.3%).

Only 8.4% of WRA tested positive for *P. falciparum* and other *Plasmodium* species. The prevalence of malaria was significantly higher in rural areas (12.8% vs. 3.5% in urban areas, p<0.001), and in the Middle Belt (12.0% vs. 5.9% in the Southern and 5.5% in the Northern Belt, p<0.05). Malaria prevalence decreased with increasing household wealth (10.4% for the poorest vs. 2.0% for the wealthiest).

The prevalence of sickle cell disease in non-pregnant women was less than 1%, but sickle cell trait was found in 13% of WRA. Similarly, the prevalence of homozygous α-thalassemia

**Table 3. Demographics, dietary diversity, infection and hemoglobinopathies among non-pregnant women of reproductive age, national, Ghana 2017.**

| Characteristic | n [a] | % [b] | (95% CI) [c] |
|---|---|---|---|
| Residing in urban household | 474 | 49.8 | (37.3; 62.4) |
| Partly or fully literate | 591 | 74.3 | (69.8; 78.3) |
| Proportion with minimum dietary diversity | 541 | 47.2 | (42.3; 52.1) |
| Any inflammation [d] | 181 | 21.0 | (17.2; 25.4) |
| No inflammation | 806 | 79.0 | (74.6; 82.8) |
| Incubation (elevated CRP only) | 65 | 6.9 | (5.5; 8.7) |
| Early convalescence (elevated CRP and AGP) | 56 | 7.0 | (4.9; 10.0) |
| Late convalescence (elevated AGP only) | 60 | 7.1 | (5.4; 9.3) |
| Malaria parasitemia | 78 | 8.4 | (5.7; 12.2) |
| Sickle cell disease or trait [e] | 60 | 13.5 | (10.6; 17.1) |
| HbSS (disease) | 2 | 0.5 | (0.1; 1.9) |
| HbAS (trait) | 58 | 13.0 | (10.1; 16.7) |
| α-thalassemia (combined) [e] | 163 | 34.6 | (29.7; 39.7) |
| Homozygous | 19 | 4.4 | (2.7; 7.2) |
| Heterozygous | 144 | 30.1 | (25.4; 35.3) |

[a] Total n was 1053 for questionnaire-related variables, 947 for malaria, 987 for inflammation, 479 for sickle cell disorders, and 474 for α-thalassemia. The n's are un-weighted numerators in each subgroup; the sum of subgroups may not equal the total because of missing data.

[b] Percentages weighted for unequal probability of selection.

[c] CI = confidence interval, calculated taking into account the complex sampling design.

[d] CRP = C-reactive protein, AGP = α1-acid-glycoprotein.

[e] Sickle cell disorders and α-thalassemia analyses were only conducted on ½ of randomly selected samples, thus the total number of samples analyzed was 479 for sickle cell disorders and 474 for thalassemia.

was low (4.4%), whereas almost one-third of the women were heterozygous for α-thalassemia. Table 4 summarizes anthropometric and micronutrient status data of WRA. While undernutrition was found in only 8.0% of the women with the majority being classified as 'at risk for undernutrition' (BMIs between 17.0–18.4), nearly one-quarter of women were overweight, and an additional 14.3% were obese. Overweight and obesity were common in Ghanaian women and increased with age (8.1% combined in 15–19 years to 61.8% in 40–44 years category, p<0.0001) and wealth quintile (16.4% combined in lowest to 58.4% in highest quintile, p<0.0001). In urban areas, almost twice the number of women were overweight and obese (49.5%) compared to rural areas with 28.8% (p<0.0001) and the Northern Belt showed the lowest prevalence (18.5% vs. 47.1% in the Southern and 41.2% in the Middle Belt, p<0.001).

Nearly 22% of WRA were anemic, with approximately two thirds of all anemia being mild. Less than 1% of non-pregnant women were severely anemic. ID and IDA affected about 14% and 9% of WRA, respectively. We found a considerable overlap of anemia and ID: 41.1% of the anemic women were also iron deficient. Anemia, ID and IDA prevalence were highest in the Northern (27.6%, 21.5% and 15.4%, respectively), medium in the Southern (23.9%, 13.6% and 9.2%, respectively), and lowest in the Middle Belt (17.5%, 10.5% and 5.9%, respectively) (p<0.05).

Only 1.5% of non-pregnant women had VAD with a higher proportion affected in the Northern Belt (4.1% vs. 1.1% in the Southern and 0.7% in the Middle Belt).

The prevalence of folate deficiency was high (>50%). Vitamin B12 deficiency was found in only 6.9% of WRA, but an additional 11.9% had a marginal vitamin B12 status. Women living in the Northern Belt had a significantly (p<0.05) higher prevalence (13.5%) of vitamin B12 deficiency compared to those in the other strata (6.3% in Middle and 3.9% in Southern Belt). The highest prevalence of marginal B12 status was found in women 45–49 years of age (35.7%) with all other age groups showing a prevalence below 18%.

Fig 2 shows different geospatial maps describing the geographical distribution of the prevalence of overweight and obesity, anemia and ID. The prevalence of overweight and obesity (Fig 2A) exceeded 30% in most EAs in the Southern and Middle Belts and was highest in the Greater Accra region, and the areas of Central and Volta regions that border Greater Accra. Regarding anemia (Fig 2B), the prevalence only exceeded 40% in some EAs in the Northern Belt. However, the interpolation shows that the anemia prevalence in women throughout Ghana is approximately in the range of 5–20%. Iron deficiency prevalence (Fig 2C) was below 20% in many areas, with a few pockets in the Northern Belt exceeding 40%.

## Discussion

Nationally, anemia in Ghanaian children and women of reproductive age indicated a moderate public health problem according to WHO classification [30], whereas for children living in the Northern Belt and children residing in rural areas the public health significance was severe (prevalence >40%). A similar disparity between the Northern Belt and the Middle and Southern Belts was observed for most of the investigated indicators. As such, the prevalence of ID and IDA in the Northern Belt was substantially higher than in the other strata. To illustrate, in children, ID was about 40% in the Northern Belt and below 20% in the Middle and Southern Belts, and IDA was more than 3 times higher in the Northern Belt compared to the other strata. Bivariate analyses showed that both ID and IDA prevalence was higher in households of the lowest wealth quintile. About 35% of total anemia in children was associated with ID (i.e. IDA), which is slightly higher than the 28% estimate in a recent meta-analysis for countries in sub-Saharan Africa [31]. This proportion was even higher in Ghanaian women (>40%), indicating that ID is an important contributor to anemia in Ghana. Thus, programs aimed at

**Table 4. Nutritional and micronutrient status among non-pregnant women of reproductive age, Ghana 2017.**

| Characteristic | National | | | Urban | | | Rural | | | p-value [c] |
|---|---|---|---|---|---|---|---|---|---|---|
| | N | % / Mean [a] | 95%CI [b] | N | %/ Mean | 95%CI | N | %/ Mean | 95%CI | |
| **Nutritional status [d]** | | | | | | | | | | |
| BMI, mean | 1003 | 24.5 | (24.1; 25.0) | 438 | 25.8 | (25.2; 26.4) | 565 | 23.3 | (22.9; 23.8) | <0.001 |
| Severely undernourished, % | 1002 | 0.5 | (0.2; 1.2) | 437 | 0.6 | (0.2; 1.9) | 565 | 0.4 | (0.1; 1.7) | 0.63 |
| Moderately undernourished, % | 1002 | 1.3 | (0.8; 2.2) | 437 | 0.9 | (0.4; 2.3) | 565 | 1.7 | (0.9; 3.1) | 0.26 |
| At risk for undernutrition, % | 1002 | 6.2 | (4.6; 8.2) | 437 | 4.3 | (2.8; 6.7) | 565 | 8.0 | (5.6; 11.2) | <0.05 |
| Normal, % | 1002 | 53.0 | (48.6; 57.4) | 437 | 44.6 | (38.9; 50.4) | 565 | 61.1 | (55.2; 66.7) | <0.001 |
| Overweight, % | 1002 | 24.7 | (21.0; 28.8) | 437 | 29.7 | (23.5; 36.9) | 565 | 19.8 | (16.2; 24.0) | <0.01 |
| Obese, % | 1002 | 14.3 | (11.5; 17.7) | 437 | 19.8 | (15.4; 25.2) | 565 | 9.0 | (6.9; 11.7) | <0.001 |
| **Micronutrient status** | | | | | | | | | | |
| Hemoglobin concentration | | | | | | | | | | |
| Hemoglobin (g/L), mean | 999 | 127.6 | (126.4; 128.9) | 436 | 127.9 | (125.8; 130.1) | 563 | 127.3 | (125.6; 128.7) | 0.62 |
| Any anemia, % | 999 | 21.7 | (18.7; 25.1) | 436 | 21.6 | (17.5; 26.4) | 563 | 21.8 | (17.4; 26.8) | 0.96 |
| Moderate anemia, % | 999 | 7.0 | (5.1; 9.5) | 436 | 8.2 | (5.7; 11.7) | 563 | 5.8 | (3.4; 9.8) | 0.29 |
| Severe anemia, % | 999 | 0.4 | (0.2; 1.3) | 436 | 0.9 | (0.3; 2.5) | 563 | - | - | 0.05 |
| Iron status | | | | | | | | | | |
| Ferritin (µg/L), median (IQR) [e] | 987 | 43.1 | (23.3; 72.4) | 438 | 40.4 | (19.2; 72.2) | 549 | 44.3 | (27.3; 72.4) | <0.05 |
| Iron deficiency, % [f] | 987 | 13.7 | (11.2; 16.6) | 438 | 15.7 | (12.5; 19.7) | 549 | 11.7 | (8.3; 16.1) | 0.15 |
| Iron deficiency anemia, % | 989 | 8.9 | (6.7; 11.7) | 435 | 9.8 | (7.2; 13.2) | 554 | 8.0 | (4.8; 12.9) | 0.48 |
| Vitamin A status | | | | | | | | | | |
| RBP (µmol/L), mean [g] | 987 | 1.64 | (1.59; 1.70) | 438 | 1.62 | (1.56; 1.68) | 549 | 1.66 | (1.57; 1.76) | 0.49 |
| Vitamin A deficiency, % [h] | 987 | 1.5 | (0.8; 2.9) | 438 | 1.1 | (0.4; 2.8) | 549 | 1.9 | (0.8; 4.4) | 0.40 |
| Folate status | | | | | | | | | | |
| Serum folate, median (IQR) | 473 | 9.3 | (5.4; 16.4) | 211 | 9.3 | (4.9; 17.0) | 262 | 9.4 | (5.5; 15.3) | 0.89 |
| Folate deficiency, % [i] | 473 | 53.8 | (47.6; 60.0) | 211 | 55.5 | (48.4; 62.3) | 262 | 52.2 | (42.3; 62.0) | 0.60 |
| Vitamin B12 status | | | | | | | | | | |
| Serum vitamin B12, mean | 471 | 454.0 | (426.8; 481.3) | 210 | 471.4 | (426.8; 516.1) | 261 | 437.0 | (402.3; 471.8) | 0.23 |
| Vitamin B12 deficiency [j], % | 471 | 6.9 | (4.8; 9.8) | 210 | 6.4 | (3.8; 10.7) | 261 | 7.4 | (4.5; 11.9) | 0.70 |
| Vitamin B12 marginal [j], % | 471 | 11.9 | (9.0;15.6) | 210 | 10.4 | (6.3; 16.7) | 261 | 13.4 | (9.9; 17.8) | 0.38 |

[a] Percentages/means weighted for unequal probability of selection.

[b] CI = confidence interval, calculated taking into account the complex sampling design. For median calculations, inter-quartile range is provided.

[c] P-values measuring the differences in mean and prevalence between urban and rural areas was calculated using t-test and chi-square test; the Mann-Whitney test was used to calculate p-values between medians.

[d] Severe undernutrition defined as BMI <16.0; moderate undernutrition defined as BMI 16.0–16.9; at risk of undernutrition defined as BMI 17.0–18.5; normal BMI defined as BMI 18.5–24.9; overweight defined as BMI 25.0–29.9; obese defined as BMI >30.

[e] Adjusted for inflammation [20]; corresponding unadjusted median ferritin concentration 45.9 µg/L (IQR: 25.0; 83.7).

[f] Based on inflammation-adjusted ferritin concentration [20].

[g] Retinol-binding protein, adjusted for inflammation [22]; corresponding unadjusted mean RBP concentration 1.65 µmol/L (95%CI: 1.59; 1.71).

[h] Based on the retinol-binding protein, adjusted for inflammation [22].

[i] Folate deficiency defined as serum folate <10 nmol/L; this analyte was measured only in a random sub-sample of women.

[j] Vitamin B12 deficiency and marginal status defined as plasma B12 <148 pmol/L & plasma B12 ≥148 & <220; this analyte was measured only in a random sub-sample of women.

reducing ID (wheat flour fortification, supplementation) should be further intensified, especially targeting poor households, the rural areas and the Northern Belt. In particular, rigorous monitoring of wheat flour fortification levels is key, as we found that only 5.7% of flour samples were adequately fortified with no difference by residence or strata [10]. However, in

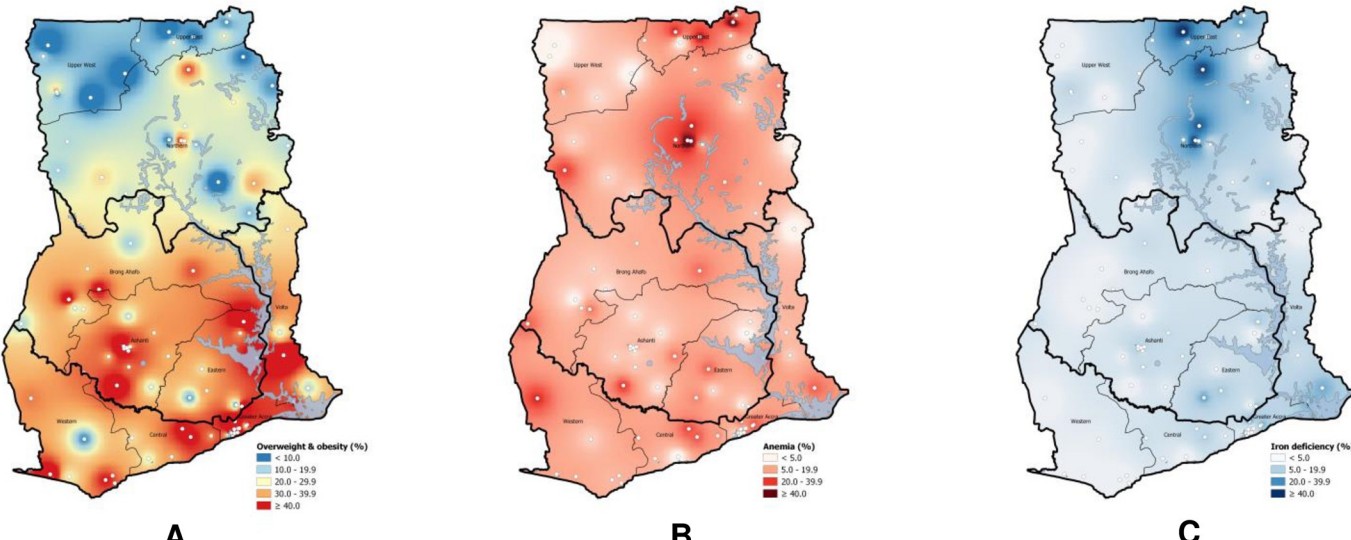

**Fig 2.** Geospatial distribution of prevalence of overweight and obesity combined (A), anemia (B), and iron deficiency (C) among non-pregnant women of reproductive age, Ghana. White dots represent the EAs.

children in the Southern Belt much less of the anemia was associated with ID than in the Northern Belt suggesting that a major cause might be infections, such as malaria for which the prevalence was twice as high in the Southern Belt as compared to the Northern Belt [10] and therefore, preventing malaria should be included in the South.

Anemia prevalence was found to be close to half in children in the GMS (35.6%) compared to the DHS in 2014 (65.7%) and half in women (21.7% vs. 42.4%, respectively). Some explanation might be the use of a different HemoCue machine (301 in GMS vs. 201+ in DHS) with the 301-model having a tendency to underestimate anemia while 201+ rather overestimates the anemia prevalence [32], venous blood samples vs. capillary blood samples (in women) which have also shown to lead to different results [32] and finally the seasonality which might have had the biggest influence as the DHS was conducted after the rainy season with a malaria prevalence of 36% vs. 20% in the GMS that was conducted just prior to the start of the rainy season and has likely resulted in lower hemoglobin values in those affected with malaria and thus a higher anemia prevalence in the DHS.

Nationally, approximately 20% of children 6–59 months were vitamin A deficient; hence VAD posed a 'severe public health problem' according to WHO classification [33], albeit the situation was only severe in the Northern Belt (>30%) and moderate with slightly below 20% in the other regions. However, when compared to the VAD prevalence of 76% in 1998 for seven regions in Ghana [34], the situation appears to have considerably improved. However, the GMS 2017 and 1998 survey results have to be compared with caution as those in 1998 were not corrected for inflammation. Furthermore, the GMS 2017 illustrated that VAD was markedly less prevalent in children residing in wealthier households, indicating that household-level socio-economic factors could influence dietary diversity, which then influences VAD in children.

Retinol-binding protein (RBP) is homeostatically controlled until liver reserves become dangerously low and is additionally influenced by the inflammatory process [35]. As such, previous reviews have recommended complementing RBP or serum retinol with the modified relative-dose response test [23]. Results of the modified relative-dose response test used during

the GMS can be found elsewhere [10], but corroborate above findings of an improvement of the VAD situation in Ghana.

Since the coverage of vitamin A supplements is low in Ghana with 28% according to our GMS 2017 data, less wealthy children likely have a higher risk of mortality due to compromised immune function [36]. Our data however also indicate that coverage rates differ by age with a higher coverage in younger children (64% and 52% in 6–11 and 12–23 month old children, respectively, as compared to 24%, 12% and 7% in 23–35, 36–47 and 48–59 month old children, respectively). This is likely due to the fact that vitamin A supplementation is linked to the child immunization schedule which is usually completed by 2 years of age. Older children are thus missed for the regular 6-monthly vitamin A supplementation. To address VAD in Ghanaian children and particularly in older children, multiple approaches should be used. First, Ghana's vitamin A supplementation program should be improved to strengthen children's immune systems and reduce the risk of mortality due to measles, diarrhea, and other illnesses. Secondly, to increase vitamin A stores, the vitamin A fortification program should be strengthened and more rigorous surveillance systems implemented to increase the coverage of adequately fortified vegetable oil. We found that overall only slightly over half of the oil samples collected in the GMS were adequately fortified with a lower proportion in the Northern Belt [10]. Thirdly, programs promoting local food products rich in provitamin A carotenoids, or introducing provitamin A-biofortified staple foods that could be readily cultivated should be pursued. This is particularly relevant in the regions where the proportion of households consuming vegetable oil is low (Upper East and Upper West).

The GMS found a very high prevalence of folate deficiency among women, with many of the women having extremely low levels below the detection limit of the laboratory method used. Such low levels of folate deficiency can result in neural tube defects and other adverse health outcomes. Systematic reviews have shown that fortification of wheat flour with folic acid [37] and folic acid supplementation are successful approaches to reduce birth defects [38]. As less than 20% of non-pregnant women were found to consume folic acid supplements in our survey, it is recommended to Ghana's health system to promote the consumption of these supplements among WRA and to initiate campaigns that raise awareness of the risk of folate deficiency and promote the consumption of folate-rich foods. Nearly 20% of women had vitamin B12 deficiency or a marginal vitamin B12 status likely a result of low animal source food consumption, particularly in the Northern Belt showing a higher prevalence. In order to prevent folate and B12 deficiencies which are likely present in children as well, the implementation of Ghana's wheat flour fortification program that already includes folic acid and vitamin B12, should be improved by ensuring that flour is fortified at adequate levels [10], as producers may add less of the premix to the flour due to perceived sensory issues [7]. Folate and vitamin B12 fortified wheat flour has shown to improve status of these two vitamins in a study in Cameroon [39].

Infectious diseases were common in children. Caregivers reported that a quarter of children had diarrhea or cough, and a third had fever during the two weeks preceding the interview. This high occurrence of illness is reflected by the high prevalence of elevated inflammation markers (AGP, CRP, or both) among the children. Eight percent of women and 20% of children tested positive for malaria by RDT, with a lower prevalence among children in urban areas and in households in the highest wealth quintile. In the DHS 2014 [40], a higher prevalence (36%) of positive malaria RDT was observed in children. These differences can be explained by the different transmission patterns as the GMS 2017 was conducted before and the DHS 2014 after the rainy season. Malaria programs to reduce exposure to malaria should be continued and strengthened, particularly targeting children from low-income households in rural areas, where malaria prevalence is highest.

Early initiation and continued breastfeeding were found to be mostly adequate, but practices with regard to minimum acceptable diet including both aspects of dietary diversity and food frequency, clearly need to be improved. A recent study has described that a minimum acceptable diet affects linear growth [41].

Child stunting affected close to 20% of the children under 5 years of age, a prevalence that is slightly lower but comparable to the 2014 DHS [40]. The prevalence was higher in rural children, and decreased with increasing household wealth. A slightly higher proportion of children in the GMS were diagnosed as wasted than in the DHS 2014 [40], 7.1% versus 4.7%. The difference in wasting may be partly explained by seasonality: the DHS 2014 was conducted following the rainy season, from September to December, when local foodstuffs are more abundant, while the GMS 2017 fieldwork was conducted at the end of the dry season, between April and May, with limited food availability. Similar to the DHS, we found a low prevalence of overweight among pre-school children with less than 1%.

Nearly 40% of non-pregnant women were classified as either overweight or obese, which compares relatively well with the 2014 DHS [40], with highest proportions found in urban areas and wealthier households, which is consistent with a recent meta-analysis [42]. There is an urgent need to prevent a further rise in overweight and obesity, which has increased by nearly 10 percentage points in the recent past, as overweight and obesity is clearly linked to type 2 diabetes, high blood pressure, cardiovascular diseases and all-cause mortality [43,44]. In particular, urban women should be educated on approaches to maintaining healthy weight, but a focus should also be on mothers as the prevalence of overweight and obesity increases with increased parity. It is therefore recommended that antenatal and postnatal care services include behavior change messages and counseling for mothers.

## Conclusions

Anemia and several micronutrient deficiencies are highly present in Ghana, especially in the Northern Belt and in poor households. Strengthening Ghana's food fortification program, ensuring particularly the coverage of rural areas and adequate fortification levels, and promoting the consumption of micronutrient rich foods might be the key to alleviate these deficiencies. This will further strengthen the immune system of young children and reduce the commonly seen illnesses. Furthermore, overweight and obesity in Ghanaian women are constantly increasing and will pose serious public health problems in the future and thus, should be addressed through governmental policies and programs.

## Acknowledgments

We thank the parents and children who participated in the survey and the fieldworkers (supervisors, interviewers, phlebotomists, anthropometrists, and drivers) who conducted the fieldwork. Others who provided particular assistance during planning, data collection and analysis include: Bradley A Woodruff, GroundWork; Jürgen G Erhardt, VitMin Lab; Lucy Twumwaah Afriyie, GSS and Setareh Shahab-Ferdows, the USDA/ARS Western Human Nutrition Research Center. We thank Alex Macharia and the staff of the Human Genetic Laboratory at the Kenya Medical Research Institute (KEMRI) in Kilifi, Kenya, for their help with genotyping. This paper is published with permission from the director of KEMRI.

## Author Contributions

**Conceptualization:** James P. Wirth, Nicolai Petry, Sherry A. Tanumihardjo, Lilian Selenje, Esi Amoaful, Matilda Steiner-Asiedu, Seth Adu-Afarwuah, Fabian Rohner.

**Data curation:** Helena Bentil, James P. Wirth, Nicolai Petry, Seth Adu-Afarwuah.

**Formal analysis:** James P. Wirth, Nicolai Petry, Seth Adu-Afarwuah, Fabian Rohner.

**Funding acquisition:** James P. Wirth, Nicolai Petry, Lilian Selenje, Abraham Mahama, Matilda Steiner-Asiedu, Seth Adu-Afarwuah, Fabian Rohner.

**Investigation:** Rita Wegmüller, Helena Bentil, James P. Wirth, Nicolai Petry, Sherry A. Tanumihardjo, Thomas N. Williams, Seth Adu-Afarwuah, Fabian Rohner.

**Methodology:** Rita Wegmüller, James P. Wirth, Nicolai Petry, Sherry A. Tanumihardjo, Lindsay Allen, Thomas N. Williams, Seth Adu-Afarwuah, Fabian Rohner.

**Project administration:** Rita Wegmüller, Helena Bentil, James P. Wirth, Seth Adu-Afarwuah, Fabian Rohner.

**Resources:** James P. Wirth, Nicolai Petry, Lilian Selenje, Abraham Mahama, Seth Adu-Afarwuah, Fabian Rohner.

**Supervision:** Rita Wegmüller, Helena Bentil, James P. Wirth, Nicolai Petry, Sherry A. Tanumihardjo, Seth Adu-Afarwuah, Fabian Rohner.

**Validation:** Rita Wegmüller, James P. Wirth.

**Visualization:** James P. Wirth, Fabian Rohner.

**Writing – original draft:** Rita Wegmüller, Nicolai Petry, Fabian Rohner.

**Writing – review & editing:** Helena Bentil, James P. Wirth, Sherry A. Tanumihardjo, Lindsay Allen, Thomas N. Williams, Lilian Selenje, Abraham Mahama, Esi Amoaful, Matilda Steiner-Asiedu, Seth Adu-Afarwuah.

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
