## [Decision Letter · Decision Letter 0]

18 Nov 2019

PONE-D-19-30143

Anemia, micronutrient deficiencies, malaria, hemoglobinopathies and malnutrition in young children and non-pregnant women in Ghana: Findings from a national survey

PLOS ONE

Dear Rita Wegmüller,

Thank you for submitting your manuscript to PLOS ONE. After careful consideration, we feel that it has merit but does not fully meet PLOS ONE’s publication criteria as it currently stands. Therefore, we invite you to submit a revised version of the manuscript that addresses the points raised during the review process.

The extent of fortification in Ghana and more details of the training quality assurance during field work.

We would appreciate receiving your revised manuscript by 17th December. To enhance the reproducibility of your results, we recommend that if applicable you deposit your laboratory protocols in protocols.io, where a protocol can be assigned its own identifier (DOI) such that it can be cited independently in the future. For instructions see: http://journals.plos.org/plosone/s/submission-guidelines#loc-laboratory-protocols

We look forward to receiving your revised manuscript.

Kind regards,

Mary Hamer Hodges

Academic Editor

PLOS ONE

Journal Requirements:

2. Please specify in your ethics statement: 1) whether the ethics committee approved the verbal/oral consent procedure, 2) why written consent could not be obtained, and 3) how verbal/oral consent was recorded.

3. We noticed you have some minor occurrence(s) of overlapping text with the following previous publication(s), which needs to be addressed:

http://groundworkhealth.org/wp-content/uploads/2018/06/UoG-GroundWork_2017-GHANA-MICRONUTRIENT-SURVEY_Final_180607.pdf

https://doi.org/10.1371/journal.pone.0155031

In your revision ensure you cite all your sources (including your own works), and quote or rephrase any duplicated text outside the Methods section. Further consideration is dependent on these concerns being addressed.

The authors have declared that no competing interests exist. The authors alone are responsible for the views expressed in this publication and they do not necessarily represent the decisions, policy or views of UNICEF.

Additional Editor Comments:

The points rasied by reviewer #1 are relevant and make the manuscript of greater interest to a global audience. The details on fortification, their reach and when these processes started will be of significant to neighboring countries

Reviewers' comments:

Reviewer's Responses to Questions

**Comments to the Author**

1. Is the manuscript technically sound, and do the data support the conclusions?

Reviewer #1: Yes

Reviewer #2: Yes

2. Has the statistical analysis been performed appropriately and rigorously? 

Reviewer #1: Yes

Reviewer #2: I Don't Know

3. Have the authors made all data underlying the findings in their manuscript fully available?

Reviewer #1: Yes

Reviewer #2: Yes

4. Is the manuscript presented in an intelligible fashion and written in standard English?

Reviewer #1: Yes

Reviewer #2: Yes

5. Review Comments to the Author

Reviewer #1: Well written paper. This is an interesting study on prevalence of anemia, B12 and folate and micronutrient deficiencies in children and women of reproductive age in Ghana. This manuscript adds to the evidence base of women at risk of obesity.

However, also consider the following minor and major corrections/suggestions to improve the manuscript.

- Background and discussion information: Information is missing on Ghana’s fortification program: Is there a fortification program? What type of program is it? Mandatory or voluntary? If there is a program what foods are fortified? With what nutrients in what concentrations? What evidence do we have of the quality and coverage of the program? Without this information, we cannot determine if there should be any expectation of better nutritional status in Ghana. For example, we would not expect a country without iron/folic acid fortification to have low anemia and/or serum folate deficiency. Similarly, comparing with other countries is not informative unless more information is presented on those countries’ fortification programs (What foods? What nutrients and concentrations? Quality and coverage of fortification?).

- It is not clear if the sample size estimation has done. If yes, prevalence indicators used? Was sample size calculation based on national or regional specificity?

- It is not documented the total duration of training?

- Similarly, standardization of anthropometry and phlebotomy trainings is also missing?

- What measures have been taken in field to avoid squeezing of finger of young children to get capillary blood sample?

Reviewer #2: It wasn't clear whether this was registered at ClinicalTrials or not?

In the section of data collection procedures, it wasn't clear what methods were used to quality control of anthropometric data

Line 195. Biological samples were shipped out of country. Was this information provided to study participants that their samples will sent out of the country?

6. PLOS authors have the option to publish the peer review history of their article (what does this mean?). If published, this will include your full peer review and any attached files.

Reviewer #1: Yes: Sajid Bashir Soofi

Reviewer #2: No

---

## [Author Response · Author response to Decision Letter 0]

4 Dec 2019

Response to reviewer’s and editor’s comments

We thank PLOS ONE for giving us the opportunity to revise the manuscript and the reviewers for their valuable comments. We have adjusted our manuscript according to the reviewers’ and editor’s comments and hope that it now fulfills the journal’s requirements. We have included a point-by-point response to each reviewer’s and editor’s comment below.

Reviewer’s comments

Reviewer 1

This is an interesting study on prevalence of anemia, B12 and folate and micronutrient deficiencies in children and women of reproductive age in Ghana. This manuscript adds to the evidence base of women at risk of obesity. 

We thank the reviewer for the encouraging words!

However, also consider the following minor and major corrections/suggestions to improve the manuscript. 

- Background and discussion information: Information is missing on Ghana’s fortification program: Is there a fortification program? What type of program is it? Mandatory or voluntary? If there is a program what foods are fortified? With what nutrients in what concentrations? What evidence do we have of the quality and coverage of the program? Without this information, we cannot determine if there should be any expectation of better nutritional status in Ghana. For example, we would not expect a country without iron/folic acid fortification to have low anemia and/or serum folate deficiency. Similarly, comparing with other countries is not informative unless more information is presented on those countries’ fortification programs (What foods? What nutrients and concentrations? Quality and coverage of fortification?). 

- We fully agree with this point. In the introduction, we already indicated which foods are fortified with which type of micronutrients. We have however extended the introduction part on food fortification in Ghana by adding the year when fortification became mandatory for the different food types, and by being more specific on the B-vitamins added to the flour as well as mentioning some results with respect to coverage of fortified foods (lines 80-87). We have also extended the fortification part in the discussion slightly by specifically mentioning the poor adherence to the fortification level for vegetable oil and in particular for wheat flour (lines 628-630, 684). 

- It is not clear if the sample size estimation has done. If yes, prevalence indicators used? Was sample size calculation based on national or regional specificity? 

A paragraph on sample size calculation has been added (line 127-133)

- It is not documented the total duration of training?

The duration has been added (line 161)

- Similarly, standardization of anthropometry and phlebotomy trainings is also missing?

Details have been added (lines 164-166)

- What measures have been taken in field to avoid squeezing of finger of young children to get capillary blood sample?

The laboratory technicians were trained on the appropriate blood collection procedure during the 10-days training prior to the survey. The pricking site, removal of the first drop, the use of the second and third drops of blood for Hb and malaria testing as well as the collection into serum tubes was standardized and technicians instructed on not to squeeze the fingers of the children. During an initial intense field supervision, this aspect was particularly looked at.

Reviewer 2

- It wasn't clear whether this was registered at ClinicalTrials or not?

No, the trial was not registered at ClinicalTrials as it is a cross-sectional survey and not a RCT. We have however registered it with the Open Science Framework study registry as stated in the Ethics section (line 110)

- In the section of data collection procedures, it wasn't clear what methods were used to quality control of anthropometric data

We have added a sentence about this aspect (lines 201-202): scales were quality controlled on a daily basis using calibration weights.

- Line 195. Biological samples were shipped out of country. Was this information provided to study participants that their samples will sent out of the country?

There was no explicit statement in the participant information sheet about samples being exported as at time of obtaining ethical clearance, this was not required by the IRB of the Ghana Health Services. However, in the survey protocol, sample export was clearly stated with the laboratories pre-identified. And as mentioned, the protocol was made public prior to implementing the survey.

Editor

The points rasied by reviewer #1 are relevant and make the manuscript of greater interest to a global audience. The details on fortification, their reach and when these processes started will be of significant to neighboring countries.

As indicated in the reviewer’s 1 section we completely agree with this point and have added the relevant information in the introduction and discussion section.

---

## [Decision Letter · Decision Letter 1]

24 Dec 2019

PONE-D-19-30143R1

Anemia, micronutrient deficiencies, malaria, hemoglobinopathies and malnutrition in young children and non-pregnant women in Ghana: Findings from a national survey

PLOS ONE

Dear Dr. Wegmüller,

Thank you for submitting your manuscript to PLOS ONE. After careful consideration, we feel that it has merit but does not fully meet PLOS ONE’s publication criteria as it currently stands. Therefore, we invite you to submit a revised version of the manuscript that addresses the points raised during the review process.

Line 71-73: the claim that “no national assessment of the prevalence of the haemoglobin disorders sickle cell disease and trait or thalassemia has previously been conducted” is wrong. At least I know that the Malawi Micronutrient Survey 2016 measured these disorders at national level. I recommend you to correct this claim and compare your findings with the finding of the aforementioned survey.In the background section, please add a paragraph that concisely present the micronutrient situation in the study country.Line 235-36: “For both PSC and WRA, ferritin concentrations were adjusted for elevated AGP and CRP according to the procedure recommended by Thurnham”. Can you please present both adjusted and non-adjusted prevalence figures in the manuscript?Line 315-16: please provide the definitions for “minimum acceptable diet”, “continued breastfeeding at 1 year” etc in the manuscript.Table 1: among many core and optional IYCF indicators it is not clear how the authors selected and presented only two of the indicators I mentioned above. I recommend them to present all the core indicators applicable to children 6-23 months in the table.It is important that the serum ferritin concentration had been adjusted for inflammation. But it is not clear why inflammation adjustments had not been made for retinol and retinol binding protein. Can you discuss the matter further? 

We would appreciate receiving your revised manuscript by Feb 07 2020 11:59PM. To enhance the reproducibility of your results, we recommend that if applicable you deposit your laboratory protocols in protocols.io, where a protocol can be assigned its own identifier (DOI) such that it can be cited independently in the future. For instructions see: http://journals.plos.org/plosone/s/submission-guidelines#loc-laboratory-protocols

We look forward to receiving your revised manuscript.

Kind regards,

Samson Gebremedhin, PhD

Academic Editor

PLOS ONE

Reviewers' comments:

Reviewer's Responses to Questions

**Comments to the Author**

1. If the authors have adequately addressed your comments raised in a previous round of review and you feel that this manuscript is now acceptable for publication, you may indicate that here to bypass the “Comments to the Author” section, enter your conflict of interest statement in the “Confidential to Editor” section, and submit your "Accept" recommendation.

Reviewer #1: All comments have been addressed

Reviewer #2: All comments have been addressed

2. Is the manuscript technically sound, and do the data support the conclusions?

Reviewer #1: Yes

Reviewer #2: Yes

3. Has the statistical analysis been performed appropriately and rigorously? 

Reviewer #1: Yes

Reviewer #2: Yes

4. Have the authors made all data underlying the findings in their manuscript fully available?

Reviewer #1: Yes

Reviewer #2: Yes

5. Is the manuscript presented in an intelligible fashion and written in standard English?

Reviewer #1: Yes

Reviewer #2: Yes

6. Review Comments to the Author

Reviewer #1: Comments are addressed appropriately, I don't have further comments on the paper. I hope it should be accepted for publication

Reviewer #2: All the comments and questions raised by the reviewers have been answered. Its a well written paper and it will add significantly to existing knowledge

7. PLOS authors have the option to publish the peer review history of their article (what does this mean?). If published, this will include your full peer review and any attached files.

Reviewer #1: Yes: Sajid Soofi

Reviewer #2: No

---

## [Author Response · Author response to Decision Letter 1]

10 Jan 2020

Response to editor’s comments

We thank PLOS ONE for re-assigning our manuscript to a new editor and appreciate the additional points raised by the new editor. We will give a point-by-point response to the additional editor’s comments below. We are glad to see that we have addressed all comments of the two initial reviewers.

Editor’s comments

Line 71-73: the claim that “no national assessment of the prevalence of the haemoglobin disorders sickle cell disease and trait or thalassemia has previously been conducted” is wrong. At least I know that the Malawi Micronutrient Survey 2016 measured these disorders at national level. I recommend you to correct this claim and compare your findings with the finding of the aforementioned survey.

In this part of the manuscript we are specifically referring to Ghana and therefore say that no national data is available. We have added ‘in Ghana’ at the end of the sentence to make this clear (line 76). There have been nationally representative assessments of hemoglobin disorders in other countries, so we hope to have clarified sufficiently that we are specifically referring to Ghana. 

In the background section, please add a paragraph that concisely present the micronutrient situation in the study country.

We have added prevalence data for those micronutrient deficiencies for which data is available in the literature (lines 70-74), which however relies on only two publications and is not national. The scarcity of the publicly available and nationally representative data was the main justification for doing the study. 

Line 235-36: “For both PSC and WRA, ferritin concentrations were adjusted for elevated AGP and CRP according to the procedure recommended by Thurnham”. Can you please present both adjusted and non-adjusted prevalence figures in the manuscript?

We prefer not to present the unadjusted prevalence figures for both, iron deficiency as well as vitamin A deficiency, as we strongly support the adjustment for inflammation in an area such as Ghana with a high infection burden. We have however added the unadjusted ferritin and RBP concentrations in the footnote of Tables 2 and 4 to give an idea on how different the adjusted values are from the unadjusted ones.

Line 315-16: please provide the definitions for “minimum acceptable diet”, “continued breastfeeding at 1 year” etc in the manuscript.

In the method section (lines 274-276) we are referring to the guidelines used for calculating the minimum acceptable diet as well as other IYCF indicators. As these are standard IYCF indicators commonly used and the focus of this paper is not on IYCF, we prefer not to give the definitions for all indicators in this manuscript to keep the manuscript focused on the biomarker topic.

Table 1: among many core and optional IYCF indicators it is not clear how the authors selected and presented only two of the indicators I mentioned above. I recommend them to present all the core indicators applicable to children 6-23 months in the table.

We agree and have included all IYCF indicators that we have assessed in the survey in Table 1 and have done some changes in the text (lines 344-345). The initial decision to only present a few selected indicators was made based on the fact that the focus of this manuscript is on biomarkers. 

It is important that the serum ferritin concentration had been adjusted for inflammation. But it is not clear why inflammation adjustments had not been made for retinol and retinol binding protein. Can you discuss the matter further? 

This is a very good point, thank you for spotting this. We have missed to state in the method section that we indeed also adjusted RBP for inflammation using Thurnham. This is now added in line 258. For both, the RBP concentration and VAD prevalence data stated in Tables 2 and 4, we used Thurnham adjusted values. This was actually stated in the footnote of Table 4 and we have added a footnote to Table 2 as well. We have additionally added a footnote in relation to iron status to also indicate that ferritin concentrations were adjusted for inflammation in both tables.

---

## [Editor Report · Decision Letter 2]

13 Jan 2020

Anemia, micronutrient deficiencies, malaria, hemoglobinopathies and malnutrition in young children and non-pregnant women in Ghana: Findings from a national survey

PONE-D-19-30143R2

Dear Dr. Wegmüller,

We are pleased to inform you that your manuscript has been judged scientifically suitable for publication and will be formally accepted for publication once it complies with all outstanding technical requirements.

With kind regards,

Samson Gebremedhin, PhD

Academic Editor

PLOS ONE

---

## [Editor Report · Acceptance letter]

17 Jan 2020

PONE-D-19-30143R2 

Anemia, micronutrient deficiencies, malaria, hemoglobinopathies and malnutrition in young children and non-pregnant women in Ghana: Findings from a national survey 

Dear Dr. Wegmüller:

I am pleased to inform you that your manuscript has been deemed suitable for publication in PLOS ONE. Congratulations! Your manuscript is now with our production department. 

With kind regards,

on behalf of

Dr. Samson Gebremedhin 

Academic Editor

PLOS ONE